# Sb Nanoparticles Embedded in the N-Doped Carbon Fibers as Binder-Free Anode for Flexible Li-Ion Batteries

**DOI:** 10.3390/nano12183093

**Published:** 2022-09-06

**Authors:** Xin Wang, Nanjun Jia, Jianwei Li, Pengbo Liu, Xinsheng Zhao, Yuxiao Lin, Changqing Sun, Wei Qin

**Affiliations:** 1School of Physics and Electronic Engineering, Jiangsu Normal University, Xuzhou 221116, China; 2Research Institute of Interdisciplinary Science and School of Materials Science and Engineering, Dongguan University of Technology, Dongguan 523820, China; 3College of Materials Science and Engineering, Changsha University of Science and Technology, Changsha 410114, China

**Keywords:** Sb nanoparticles, N-doped carbon fibers, free-standing anode, lithium-ion battery

## Abstract

Antimony (Sb) is considered a promising anode for Li-ion batteries (LIBs) because of its high theoretical specific capacity and safe Li-ion insertion potential; however, the LIBs suffer from dramatic volume variation. The volume expansion results in unstable electrode/electrolyte interphase and active material exfoliation during lithiation and delithiation processes. Designing flexible free-standing electrodes can effectively inhibit the exfoliation of the electrode materials from the current collector. However, the generally adopted methods for preparing flexible free-standing electrodes are complex and high cost. To address these issues, we report the synthesis of a unique Sb nanoparticle@N-doped porous carbon fiber structure as a free-standing electrode via an electrospinning method and surface passivation. Such a hierarchical structure possesses a robust framework with rich voids and a stable solid electrolyte interphase (SEI) film, which can well accommodate the mechanical strain and avoid electrode cracks and pulverization during lithiation/delithiation processes. When evaluated as an anode for LIBs, the as-prepared nanoarchitectures exhibited a high initial reversible capacity (675 mAh g^−1^) and good cyclability (480 mAh g^−1^ after 300 cycles at a current density of 400 mA g^−1^), along with a superior rate capability (420 mA h g^−1^ at 1 A g^−1^). This work could offer a simple, effective, and efficient approach to improve flexible and free-standing alloy-based anode materials for high performance Li-ion batteries.

## 1. Introduction

With the increasing aggravation of environmental pollution and demand for energy conversion and storage, it is critical to develop advanced energy storage devices nowadays [1]. Among the existing energy storage devices, Li-ion batteries (LIBs) have many advantages such as high energy density and high operating voltage; and are extensively used as power sources in various low power electronic devices such as mobile phones, laptops, electric bicycle, and other small power devices [2,3]. However, the commercial LIBs use graphite with low theoretical specific capacity (372 mA⋅h⋅g^−^^1^) as the anode active materials, which greatly restrict their wide application in electric vehicles and the smart-grid where high power densities are needed. Therefore, it is important to search for alternative anode materials with high specific capacity and good rate capability. So far, various kinds of anode materials such as metal materials [4,5,6,7], semimetal materials [8], transition metal oxides [9,10,11,12,13], metal chalcogenides and transition metal chalcogenides have been studied extensively [14,15,16]. Among them, Sb as a metal material is recognized as a promising anode material for LIBs because of its high theoretical specific capacity (660 mAh g^−1^), fast charge carrier mobility, low electronegativity, decent working voltage, puckered-layer architecture and good thermodynamic properties [17]. Unfortunately, the bulk Sb electrode is primarily hampered by its gigantic volume change (~150%) during the alloying/dealloying reaction with Li, which would cause structural pulverization, and then result in a rapid capacity fade of the cycling performance for bulk Sb anode materials [18].

It is known that nanomaterials have better mechanical properties and a larger specific area compared with those of corresponding bulk materials [19,20,21]. As a result, when the size of materials reduces to nanoscale, not only can the volume expansion of the active materials in the process of lithium intercalation be inhibited, but also the transmission path of the lithium ion in the electrode material may be shortened [6]. Therefore, reducing the dimension of Sb to nanoscale is an effective strategy to alleviate volume change. For example, Meng He et al. reported that monodisperse Sb nanoparticles showed good capability as anode materials in rechargeable Li-ion after 100 cycles. Nevertheless, the longer-term cycling ability is still limited due to the aggregation of Sb nanoparticles and the unstable SEI film [18]. To address these problems, different kinds of Sb/carbon composites have been synthesized. For example, Sb/carbon nanocomposites were synthesized using high energy ball milling by Glushenkov et al. [22], which exhibited a good reversible capacity of 550 mAh g^−1^ at a current rate of 230 mA g^−1^ after 250 cycles. Qian et al. prepared rod-like Sb/C composites by a synchronous reduction and carbon deposition process, which exhibited a reversible capacity of 478.8 mAh g^−1^ at 100 mA⋅g^−1^ after 100 cycles [23]. Thus, it is recognized that constructing Sb/C nanoscale composites can successfully improve the cycling ability of the Sb material. 

Compared with 3D and 2D materials, the ionic diffusion and electronic transport length can be effectively shortened along their geometry in 1D materials, leading to fast diffusion kinetics. Electrospinning is a feasible technique to prepare various 1D carbon-containing composites. The physical and chemical properties, such as surface-to-volume ratio, mechanical strength, crystallinity, particle size and distribution of metal active materials can be tuned by using different electrospinning precursor solutions and processing parameters [24,25,26]. A series of anode materials have been prepared by electrospinning such as Sn/NPCFs [7], Ge/CNFs [27], Si/C@CNFs and Fe_2_O_3_ @CNFs [9,28]. This method can also be adopted to prepare flexible and free-standing films [29,30] as working electrodes of rechargeable batteries without carbon black or binder additives [31], which could improve the volumetric energy and power density of batteries and are beneficial for the overall electrochemical performance. 

Considerable efforts have been made to design highly flexible and binder-free electrodes by thermal oxidation [32], electrodeposition [33], vacuum filtration [34], self-assembly [35,36], hydrothermal method [37], and electrospinning [38,39]. However, it should be noted that forming intimate contacts between active materials and freestanding matrices via the construction of a flexible and free-standing electrode for LIBs is still a great challenge. For most freestanding electrodes, the active materials are only physically interconnected with the freestanding matrices without stable interfaces. Consequently, large stress will be generated during the repeated cycling test, resulting in poor cycling stability of the batteries. Therefore, constructing freestanding electrodes with a stable interface is important for the practical application of freestanding electrodes in flexible LIBs.

Herein, for the first time, combining the advantages of Sb nanoparticle and 1D carbon fibers, we report the synthesis of free-standing Sb nanoparticles embedded in 1D flexible N-doped carbon fibers (denoted as Sb@NCFs) using an electrospinning method for LIBs. Owing to the robust hierarchical structure and shortened Li-ion diffusion distance, the as-prepared Sb@NCFs exhibit excellent Li-ion storage performance including high specific capacity, superior rate performance and stable cycling stability. 

## 2. Materials and Methods

### 2.1. Materials

Polyacrylonitrile (PAN, Mw = 1,500,000) and polyvinylpyrrolidone (PVP, Mw = 1,300,000) were purchased from Sigma-Aldrich (Shanghai) Trading Co., Ltd., Shanghai, China. Diisopropanolamine (DIPA) was purchased from Alfa Aesar (China) Chemicals Co., Ltd., Shanghai, China. N, N-Dimethylformamide (DMF) and antimony (Ⅲ) chloride (SbCl_3_) were from Sinopharm Chemical Reagent Co., Ltd., Shanghai, China. The reagents above were of analytical purity grade and used without further purification.

### 2.2. Synthesis of Sb@NCFs 

The preparation process of Sb@NCFs is shown in Figure 1. The free-standing Sb@NCFs was synthesized by the electrospinning technique as follows: firstly, a certain amount of PAN (0.75 g) and PVP (0.25 g) were dissolved into 10 mL DMF, followed by adding 0.3 mL DIPA. The mixture above was continuously stirred for 12 h at room temperature to obtain a homogeneous solution. Subsequently, the solution was mixed with 0.7 g SbCl_3_ under continuous vigorous stirring for 3 h. Lastly, the obtained precursor solution was loaded into a 20 mL plastic syringe equipped with a 21# stainless steel needle for electrospinning. A sheet of copper foil on a stainless-steel drum was used as the collector and placed 15 cm away from the tip of the stainless-steel needle. The spinning voltage, the rotating speed of the drum, and the injection flow rate were 15 kV, 60 RPM, and 0.018 mm/min, respectively. Then, the as-spun fibers were placed between the ceramic sheets and annealed at 250 °C for 2 h at a heating rate of 3 °C/min in air. Finally, the as-spun fibers were annealed at 600 °C for 2 h at a heating rate of 2 °C under N_2_ atmosphere to obtain the Sb@NCFs (denoted as Sb_3_). To investigate the effect of organic carbon source precursor on the electrochemical properties of Sb@NCFs, the experiment without DIPA in precursor solution was carried out and the as-synthesized sample denoted as Sb_2_, the sample synthesized without PVP and DIPA in precursor solution (1.0 g PAN) was denoted as Sb_1_.

### 2.3. Material Characterizations 

The phase and composition of Sb@NCFs were analyzed using X-ray diffraction (XRD, Bruker D8, Karlsruhe, Germany) techniques with Cu-Ka radiation (λ = 0.1542 nm, V = 40 kV, I = 40 mA) at scanning rate of 0.05° s^−1^ in the range of 10–80°. The morphologies of the Sb@NCFs were characterized using scanning electron microscope (SEM, JSM-6510, JEOL, Tokyo, Japan) and field emission transmission electron microscopy (FETEM, Tecnai G2 20, OR, USA). X-ray photoelectron spectroscopy (XPS, Thermo ESCALAB 250XI XPS, MA, USA) was employed to analyze compositions and chemical states of the elements. The specific surface areas and pore size distributions were recorded using the Brunauer–Emmett–Teller (BET) surface area analysis (Autosorb-IQ), FL, USA. Thermogravimetric analysis (TA) was performed in air from 25–800 °C with a heating rate of 5 °C min^−1^ by using an TA thermogravimetry analyzer (Q500), Delaware, FL, USA.

### 2.4. Electrochemical Characterizations

The electrochemical tests were performed on CR2032 coin cells. First, the Sb@NCFs were directly cut into circular electrodes (8 mm in diameter, 0.15~0.45 mg in weight) and used as working electrode. Then, coin cells were assembled in an argon-filled glove box (MBraun, Munich, Germany), of which the working electrode, separator, counter and reference electrode were Sb@NCFs, Celgard 3501 and lithium foil with an electrolyte of 40 μL LiPF_6_ in a 4:5:1 (*v*/*v*) EC/DEC/PC, respectively. After standing for 10–12 h with the filled electrolyte, the assembled cells were connected to the Arbin battery test system (BT2000, Arbin Instruments, College Station, TX, USA). The galvanostatic discharge–charge and rate performance were conducted in the voltage range from 0.01 to 2.8 V at a room temperature. Cyclic voltammetry (CV) curves were obtained on an electrochemical workstation (CHI 760E, Shanghai, China) in the potential range of 0.01–2.8 V (vs. Li^+^/Li) at a scan rate of 0.1 mV s^−1^. The electrochemical impedance spectroscopy (EIS) was measured in the frequency range of 10^−1^ to 10^6^ Hz with an AC voltage amplitude of 5 mV.

## 3. Results and Discussion

### 3.1. Characterizations of Sb@NCFs Composite 

The crystal structures of the Sb_1_, Sb_2_ and Sb_3_ were characterized using XRD and are shown in Figure 2a. All the diffraction peaks could be indexed to Sb phase (JCPDS 35-0732) [40,41]. The main diffraction peaks of Sb located at 28.69°, 40.077°, and 41.947° corresponded to the (012), (104), and (110) planes. No impurities could be detected, which implied Sb^3+^ was reduced completely to metallic Sb by carbothermal reduction reaction [42,43]. Moreover, one could see that by adding PVP and DIPA in the precursor, Sb_3_ exhibited better crystallinity than that of Sb_1_ and Sb_2_.

To investigate the morphologies of Sb_1_, Sb_2_ and Sb_3_, the SEM images are presented in Figure 2b and Appendix A. As can be seen in Figure 2b, the Sb_3_ was composed of long and cross-linked fibers with smooth surfaces and with diameters ranging from 300 to 400 nm. The morphologies of Sb_2_ and Sb_3_ were similar to Sb_3_ (Appendix A). This unique one-dimensional nanostructure can facilitate the rapid transfer of electrons and Li^+^. As shown in Figure 2c–e, the TEM of Sb_3_ was performed to check whether the metallic Sb had good contact with N doped carbon fibers. The TEM images displayed smooth surfaces without Sb particles, indicating that all Sb particles were encapsulated in the N-doped carbon fibers without agglomeration, which was consistent with the results of SEM. The conductive N-doped carbon coated on the outside of the metal particles not only served as a support but also suppressed the volume change during insertion/removal of lithium, further increasing the specific capacity of the Sb_3_. The HRTEM image in Figure 2e shows the lattice fringe of 0.21 nm, which was consistent with the XRD results and corresponded to the (110) plane of Sb with hexagonal phase (JCPDS: 35-0732). To further analyze the element distribution of the Sb, C and N in the Sb_3_, EDS was carried out. As shown in Figure 2g–j, all the energy-filtered SEM maps of the Sb, C and N matched well with the SEM image, indicating that the Sb, C and N were uniformly dispersed and coexist over the entire composite. These features are beneficial to relieve volume expansion and increase electrical conductivity when used as electrode materials of batteries. The as-spun Sb@NCFs showed excellent flexibility, as demonstrated by the optical image shown in Figure 2f, which could thus be used as binder-free electrode of LIBs directly. The XPS spectra were also employed to distinguish the chemical composition and valence state of Sb_3_ in Figure 3a. The peaks of Sb 3d, Sb 4d, C 1s and N 1s were observed in the survey scan spectra, indicating the coexistence of Sb, C, and N elements [44]. The C 1s spectrum is given in Figure 3b. The peaks at the binding energy of 284.6, 286.0, 287.0 and 288.2 eV were assigned to the binding energies of graphitized carbon, C-N group, C=C group, and O=C-N group bonds, respectively [45]. The peaks at the binding energy of 398.3 eV, 400.1 eV, 401.2 eV, and 402.7 eV were assigned to pyridine N, pyrrole N, quaternary N, and oxidized N (Figure 3c), respectively [46]. It is beneficial for the insertion of lithium ions because pyridine N and pyrrole N create more open channels and active sites [47]. The binding energies at 530.8 and 540.1eV were assigned to Sb 3d_5/2_ and Sb 3d_3/2_ (Figure 3d), respectively [48]. There was a peak at ca. 199 eV, which is the fingerprint peak of Cl [49], meaning that the Cl element from SbCl_3_ had been completely removed in the carbonization process.

The as-synthesized free-standing films were ground into powders to investigate the pore structures. The N_2_ adsorption–desorption isotherms of the Sb_1_, Sb_2_ and Sb_3_ were carried out and are presented in Appendix A. The adsorption–desorption isotherms exhibited the characteristics of type-IV isotherm with a H_3_ hysteresis loop (Appendix A), indicating the mesoporous structure of the Sb_1_, Sb_2_ and Sb_3_ [41]. The specific surface areas of Sb_1_, Sb_2_, Sb_3_ were 0.632, 5.193, and 24.06 m^2^ g^−1^, respectively. The pore-size distributions were calculated according to density functional theory and are shown in Appendix A, indicating Sb_3_ had a hierarchical porous structure with abundant mesopores and a small number of micropores centered at 1.4 nm. Correspondingly, the total pore volume of Sb_3_ was 0.099 cm^3^ g^−1^ while those of Sb_1_ and Sb_2_ were only 0.022 cm^3^ g^−1^ and 0.006 cm^3^ g^−1^. The large specific surface area and pore volume of Sb_3_ will contribute to the electrode/electrolyte contact area, providing sufficient active sites for charge transfer, shortening the diffusion length of the Li ions and finally enhancing the electrochemical performance.

The thermogravimetric (TG) curve of the Sb_1_, Sb_2_ and Sb_3_ (Appendix A) was conducted to determine the content of Sb in the Sb@NCFs composite. The slight weight loss below 300 °C was mainly associated with the evaporation of water absorbed on the surface of samples. The weight loss between 300 °C and 600 °C can be attributed to the burnt out of carbon and oxidation of Sb particles. The residual part of Sb_3_ was around 22.9 wt.%, while Sb_2_ and Sb_1_ were around 15.5 wt.% and 10.7 wt.% after weight loss. The results showed that the Sb_3_ had the highest Sb content of all the samples. 

### 3.2. Electrochemical Performance of Sb@NCFs Composite 

The Sb@NCF electrode was mechanically flexible, as shown in Figure 1f. The selected first, second, and fifth discharge/charge curves of Sb_1_, Sb_2_ and Sb_3_ between 0.01 and 2.8 V at a current density of 0.1 A g^−1^ are shown in Figure 4. The first discharge curve of Sb_3_ (Figure 4c) showed that the voltage dropped rapidly to 0.8 V, which corresponded to the formation of the solid electrolyte interphase (SEI) on the electrode surface [42]. Then, a discernable plateau at about 0.8 V appeared which indicates the formation of a Li_3_Sb phase during lithium insertion. As the voltage decreased to 0.01 V, the battery exhibited an initial specific capacity of 675.2 mAh g^−1^, higher than the theoretical specific capacity of Sb (660 mAh g^−1^), which could be attributed to irreversible Li ion consumption owing to the decomposition of the electrolyte and the formation of SEI films [50]. The first charge curve of Sb_3_ exhibited a plateau of 1.0 V which was associated with the reversible dealloying process, with a specific capacity of 646.9 mAh g^−1^. For Sb_2_, the first discharge/charge curves (Figure 4b) were similar to the curves of Sb_1_ (Figure 4a). The initial charge and discharge specific capacities of Sb_2_ and Sb_1_ were 529.4, 568.7, 474.7 and 496.1 mAh g^−1^, respectively. Compared with Sb_2_ and Sb_1_, Sb_3_ delivered higher specific capacities which could be ascribed to its higher specific surface area, mesoporous structure, and content of Sb. 

The initial three CV curves of Sb_3_ at a scan rate of 0.1 mV s^−1^ between 0 and 2.8 V (vs. Li^+^/Li) are presented in Figure 5a. In the first cathodic sweep, there was a strong cathode peak at about 0.3 V which was attributed to the formation of SEI film [6,42]. The peak did not appear in the second cycle, which reflects the high stability of the SEI film. The cathodic peak was located at 0.8 V in following cycles, which was ascribed to the reaction of metallic Sb to alloyed Li_3_Sb [6]. Apart from the first cycle, the CV curves overlapped well in the following cycles, demonstrating the good electrochemical reversibility of Sb@NCFs during cycling. Based on the analysis above, the electrochemical process for Sb@NCFs is as follows [51]:(1)$$Sb+3Li^{+}+3e^{-}↔Li_{3}Sb$$

To further investigate the electrochemical Li-ion storage performance at high current densities, the rate capabilities of Sb_1_, Sb_2_ and Sb_3_ are displayed in Figure 5b. As seen, the Sb_3_ sample exhibited the reversible capacities of ca. 620, 573.9, 492.5, 459.7, 441.8 and 421.5 mAh g^−1^ at the current densities of 0.1, 0.2, 0.4, 0.6, 0.8 and 1 A g^−1^, respectively. It was noted that the discharge capacity of Sb_3_ returned to 585.3 mAh g^−1^ when the current density recovered to 0.1 A g^−1^, suggesting that the structure of Sb_3_ electrode keeps integrity after high current charge–discharge process [22,52]. The specific capacities (ca. 530.4, 456, 386.4, 344.4, 333.3, and 323.6 mAh g^−1^) of Sb_2_ at the current densities of 0.1, 0.2, 0.4, 0.6, 0.8, and 1 A g^−1^ were lower than those of Sb_3_ without DIPA, owing to the smaller pore size and specific surface area. For Sb_1_ without DIPA and PVP, the specific capacities (ca. 496.1, 220.6, 180.9, 149.2, 125.5, and 104.3 mAh g^−1^) were even lower than those of Sb_2_ and Sb_3_. When the current density returned to 0.1 A g^−1^, the discharge capacity was only 460.7 mAh g^−1^ for Sb_2_ and 243.1 mAh g^−1^ for Sb_1_. The cycling stability of the Sb@NCFs electrode at a current density of 0.1 and 0.4 A g^−1^ is shown in Figure 5c. The discharge capacity decayed to 629 mAh g^−1^ after the first cycle and a high reversible discharge capacity of ca. 590 mAh g^−1^ could still be retained with a high coulombic efficiency after 300 cycles. Even at a current density of 0.4 A g^−1^, the Sb@NCFs electrode still maintained a high specific charge capacity of 480 mAh g^−1^ electrode after 300 cycles.

To further clarify the electrochemical performance of the Sb@NCFs, EIS measurements were performed after the 10 cycles (Figure 6). All Nyquist plots can be divided into three parts, the Z’-axis intercepts, the semicircle, and the oblique line in the high and low-frequency region, which represent the resistance of electrolyte (R_e_), the resistance of charge transfer (R_ct_), and the Warburg (Z_w_) impedance related to the lithium-ion diffusion (D_Li_), respectively [23,44,53,54]. The Nyquist plots were fitted with the equivalent electric circuit and the values of the R_e_, R_ct_ and Z_w_ are shown in the inset table of Figure 6. The diameter of the high-frequency semicircle revealed that the Sb_3_ had the lowest R_ct_ value, which indicates the reduction of the unfavorable side reactions on the electrode/electrolyte interface for Sb_3_. The slope value of the low-frequency oblique line showed a trend of Sb_3_ > Sb_2_ > Sb_1_, which indicated that the Sb_3_ had the largest D_Li_ and fastest lithium intercalation/deintercalation, since Sb_3_ had the highest conductivity and porosity. It could thus be concluded that Sb_3_ had the lowest impedance (including R_e_, R_ct_, and Z_w_) at the electrode/electrolyte interface, leading to high specific capacity retention and a superior rate performance [55,56,57,58].

Compared with the results reported in the literature, our Sb@NCF composite exhibited better electrochemical properties, as listed in Table 1 [6,22,23,42,44,52,59,60]. Considering all the above results, the Sb@NCF composite is more suitable for anode material than other developed materials for future LIBs. First, there is no need to add conductive agents, binders and collectors in the electrode preparation process. The flexible free-standing electrode is cost effective with excellent electrochemical performance. Secondly, the carbon matrix can prevent Sb nanoparticles from agglomerating into bulk particles and suppressing the volume change during repeated charging and discharging processes. Finally, the porous structure can provide rich electrolyte penetration channels and additional space to relieve structural decomposition/extraction caused by lithium-ion insertion/removal. Thus, the rate capability and cycling stability is significantly improved.

## 4. Conclusions

In summary, a highly flexible Sb@NCF composite with excellent electrochemical performance was prepared by the electrospinning method. The Sb nanoparticles were uniformly dispersed in the carbon matrix which could provide a stable porous structure for the rapid charge transfer and Li-ion intercalation/deintercalation. When used as a free-standing electrode of the LIB anode, the unique Sb@NCF electrode delivered a high reversible capacity of 675 mAh g^−1^ at the current density of 100 mA g^−1^ and excellent capacity retention rates of 88 and 87.6% at a current density rate of 100 and 400 mA g^−1^ after 300 cycles, respectively. The new flexible and binder-free electrode enables the practical applications of Sb-based anodes in high energy density LIBs.

## Figures and Tables

**Figure 1 nanomaterials-12-03093-f001:**
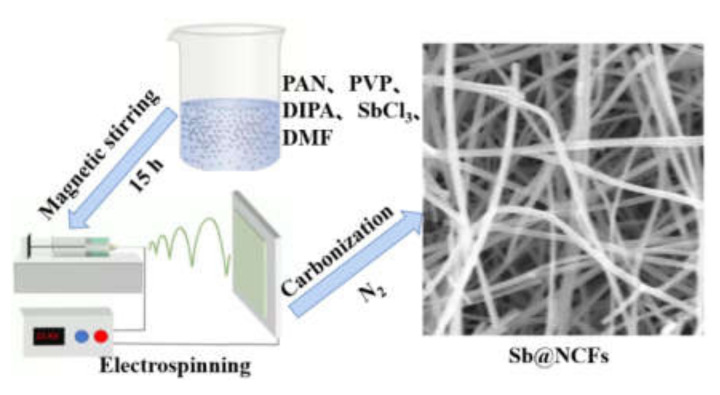
Synthetic illustration of the synthetic process of Sb@NCFs.

**Figure 2 nanomaterials-12-03093-f002:**
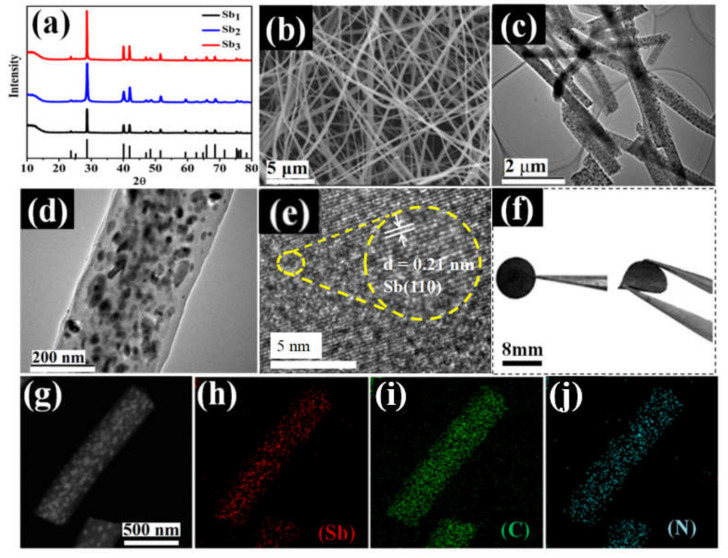
(**a**) XRD patterns of Sb_1_, Sb_2_, Sb_3_, (**b**) SEM images of Sb_3_, (**c**,**d**) TEM image of Sb_3_, (**e**) HRTEM image of Sb_3_, (**f**) optical images of flexible Sb@NCFs, and (**g**) HAADF-STEM image of Sb_3_ and EDS mappings of (**h**) Sb, (**i**) C and (**j**) N.

**Figure 3 nanomaterials-12-03093-f003:**
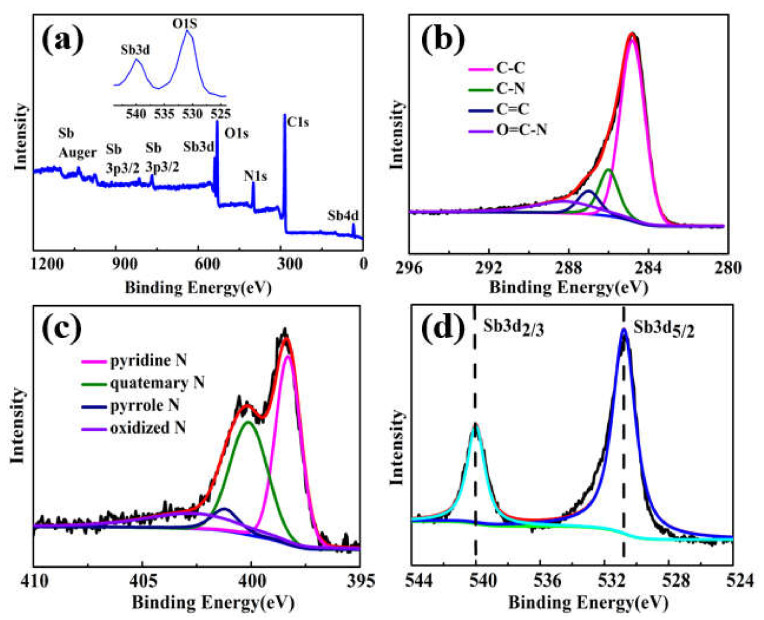
XPS spectra of Sb_3_: (**a**) survey scan, (**b**) C1s, (**c**) N1s, (**d**) Sb3d.

**Figure 4 nanomaterials-12-03093-f004:**
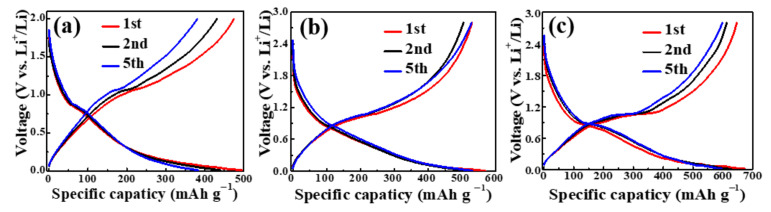
Galvanostatic discharge/charge curves of the 1st, 2nd, 5th cycles for (**a**) Sb_1_, (**b**) Sb_2_ and (**c**) Sb_3_ at a current density of 100 mA g^−1^.

**Figure 5 nanomaterials-12-03093-f005:**
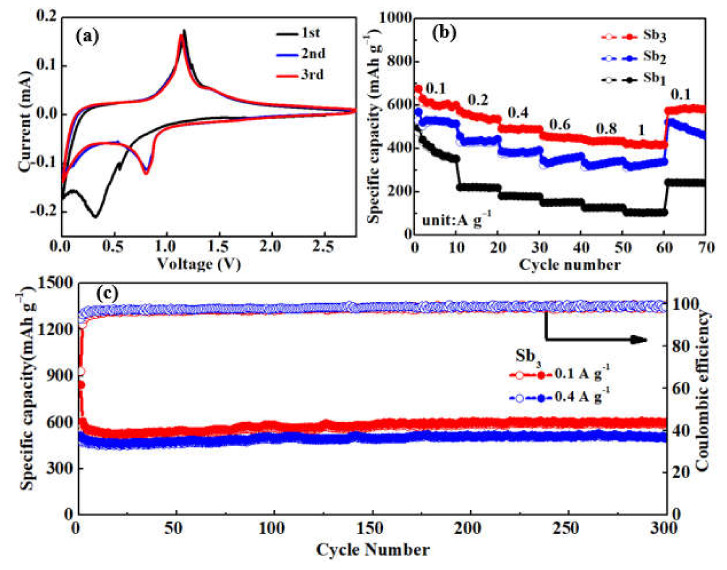
(**a**) CV curves of Sb_3_, (**b**) rate performances of Sb_1_, Sb_2_, Sb_3_, galvanostatic discharge–charge cycles of Sb_3_ at a current density of (**c**) 0.1 Ag^−1^ and 0.4 Ag^−1^.

**Figure 6 nanomaterials-12-03093-f006:**
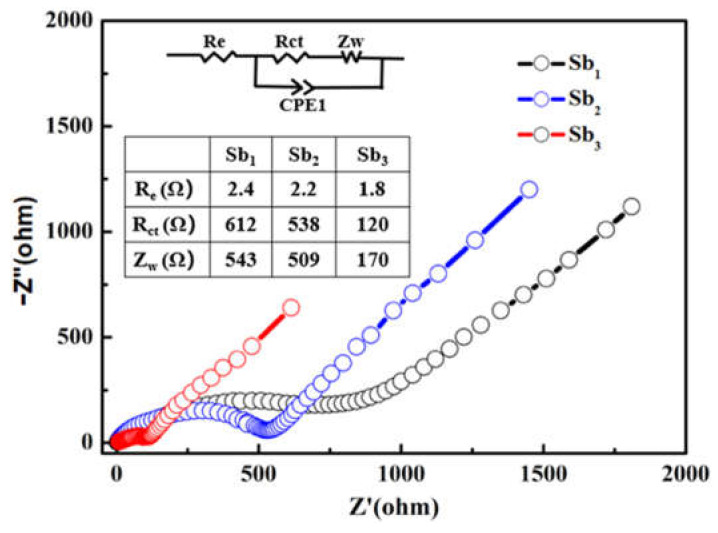
Nyquist plots of the Sb_1_, Sb_2_ and Sb_3_.

**Table 1 nanomaterials-12-03093-t001:** Comparison of the electrochemical performances of the prepared Sb@NCF composites with previously reported Sb-C-based composites.

Materials	Free-Standing (Yes or No)	Current Density (A g^−1^)	Cycle Numbers	Capacity (mAh g^−1^)	References
Sb/CNT	No	0.05	30	287	[59]
Sb@C	No	0.3	20	408	[52]
Sb/C	No	0.23	250	550	[22]
		1.15	-	400	
Sb/CNT	No	0.25	50	277.4	[60]
Sb/C	No	0.1	100	315.9	[42]
		0.8	-	246.2	
Sb@C	No	0.1	100	478.8	[23]
		0.5	-	369.7	
Sb/graphite	Yes	0.1	50	424.1	[44]
		1	-	158	
Sb/C	No	0.2	100	565	[6]
		1	500	400.5	
		5	-	315.4	
Sb@NCFs	Yes	0.1	300	590	This work
		0.4	300	480	

## Data Availability

The data presented in this study are available on request from the corresponding author.

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
