# Peer review of "Sb Nanoparticles Embedded in the N-Doped Carbon Fibers as Binder-Free Anode for Flexible Li-Ion Batteries"

_nanomaterials, 2022, doi:10.3390/nano12183093_

Round 1

Reviewer 1 Report

I have read the article “Sb Nanoparticles Embedded in the N-doped Carbon Fibers as Binder-free Anode for Flexible Li-ion Batteries” by Wei Qin submitted to Nanomaterials, MDPI. Among the available energy storage technologies, lithium-ion technology is an attractive energy storage device and dominates over lead-acid and sodium-ion. This is because Li-ion exhibits excellent efficiency in charge-discharge processes along with high energy density. They are an indispensable power source for a variety of portable electronic devices and EVs today. Among the available anodes, Sb has been considered as a promising anode material due to its high theoretical capacity (660 mAh g−1) and low reaction potential. However, the large volumetric change during charging/discharging leads to a sharp decay of its capacity. Therefore, in the submitted work, the author proposed a hybrid form of Sb nanoparticles/N-doped carbon fibers, this could provide more active sites for ion insertion/extraction.

The work is well presented, and the manuscript is OK with reasonable physical and electrochemical data drawn from the morphological images and surface properties. The work in its present form is publishable but needs some revisions before rendering a final decision. 

The following points need to be considered.

·         LTO and Si-based materials are also excellent anode materials used in Li-ion batteries, which needs to be included in the introduction.

·         Ge et al (Angew Chem Int Ed 58(41):14578–14583] synthesized ultra-small Sb nanocrystals within carbon nanofibers containing hollow nanochannels (u-Sb@CNFs) through electrospinning; and used an anode, delivering a reversible capacity of 225 mA∙h/g over 2000 cycles at 1 A/g. How does the current work compare to those of reported values? Likewise, a self-wrapped Sb/C nanocomposite was reported in Nano Energy (2015) 16:479. Please discuss.

·         Please provide the alloying/dealloying reaction involved during the redox process. Will the different configurations of Sb-Carbon affect the storage properties?

·         In the introduction section (related to NCM and cathodes for Li-ion batteries), relevant reported works on alternative materials illustrating potential candidates expressed in these works (such as Progress in Solid State Chemistry 62 (2021) 100298; Energies 13 (2020) 1477) needs to be included and discussed.

·         In addition to using carbon to coat Sb for enhancing the performance of lithium storage, compositing Sb with other stable materials in the charge and discharge processes is another effective approach been widely researched. Any comment?

·         In Figure 6, please provide the Re, Rs, and Rct values.

·         What is the role of DIPA? Is that result in enhancing the conductivity?

·         Please compare the results of the well-known anode NiMoO4 reported in the literature with Sn anode.

·         Are the functional groups of Nitrogen (N) seen in XPS act as composites (or heteroatom dopants) in the anode? To improve the ion transport kinetics and electron diffusion ability of Sb active materials, thereby providing the possibility to obtain a high-rate performance. Please justify.

Reviewer 2 Report

The manuscript requires a thorough grammar and spell check.

Round 2

Reviewer 1 Report

In this reviewer's opinion, the authors have revised the manuscript suitably.